# Iterative Online 3D Reconstruction from RGB Images

**DOI:** 10.3390/s22249782

**Published:** 2022-12-13

**Authors:** Thorsten Cardoen, Sam Leroux, Pieter Simoens

**Affiliations:** IDLab, Department of Information and Technology, Ghent University-imec, 9052 Ghent, Belgium

**Keywords:** 3D reconstruction, edge computing, deep learning

## Abstract

3D reconstruction is the computer vision task of reconstructing the 3D shape of an object from multiple 2D images. Most existing algorithms for this task are designed for offline settings, producing a single reconstruction from a batch of images taken from diverse viewpoints. Alongside reconstruction accuracy, additional considerations arise when 3D reconstructions are used in real-time processing pipelines for applications such as robot navigation or manipulation. In these cases, an accurate 3D reconstruction is already required while the data gathering is still in progress. In this paper, we demonstrate how existing batch-based reconstruction algorithms lead to suboptimal reconstruction quality when used for online, iterative 3D reconstruction and propose appropriate modifications to the existing Pix2Vox++ architecture. When additional viewpoints become available at a high rate, e.g., from a camera mounted on a drone, selecting the most informative viewpoints is important in order to mitigate long term memory loss and to reduce the computational footprint. We present qualitative and quantitative results on the optimal selection of viewpoints and show that state-of-the-art reconstruction quality is already obtained with elementary selection algorithms.

## 1. Introduction

3D reconstruction is the task of recovering the 3D geometries of objects from multiple images that were captured from different viewpoints. Multi-view 3D reconstruction has applications in offline tasks such as medical imaging [1] and object recognition [2], but is also a building block of real-time processing pipelines for robot navigation, robot grasping and robot situational awareness. In the first category of tasks, which we refer to as batch-based reconstruction, the reconstruction algorithm is executed after all images have been captured, and the sole performance metric is the reconstruction accuracy. The second category contains mostly robotic control tasks where an accurate reconstruction is needed simultaneously with the data gathering, a setting we refer to as iterative 3D reconstruction. Multiple reconstructions are performed and potentially refined as additional viewpoints become available, e.g., as the robot moves.

Reconstruction algorithms have been presented in the literature that operate on various sensor modalities, including LIDAR [3], RADAR [4], RGB-D [5] and monocular (RGB) cameras [6,7,8,9,10,11,12,13]. In this paper, we focus on reconstruction from RGB images, since cameras are an attractive sensor modality for aerial and ground robots because they are more affordable, lighter and power-efficient than active sensor modalities. Moreover, monocular perception and an inertial measurement unit form the minimum sensor layout for sufficient self and environmental perception of autonomous agents [14,15].

Early methods for 3D reconstruction focused on the mathematical and geometric aspects of reconstruction. Viewpoints are interpreted as 2D projections, and the goal is to determine the inverse projection from matching features across viewpoints. To make this feature matching tractable, multiple cameras are needed with largely overlapping viewports and within a well-calibrated distance. With the advent of deep learning, scholars have been able to frame the problem of reconstruction as a recognition problem [16]. Deep neural networks (DNNs) are trained on datasets containing 3D shapes with associated 2D RGB images, with the R2N2 [6] dataset being the most widely used benchmark. Existing literature has predominantly studied DNN architectures for batch-based reconstruction. No spatial or temporal ordering in the batch is assumed. The batch is sampled from a limited set of spatially distant viewpoints in the dataset, and performance is only measured in terms of the one-shot reconstruction quality. In online, iterative 3D reconstruction, illustrated in Figure 1, these assumptions on the composition of the input batch do not hold. Moreover, the goal is to maintain at all times a good reconstruction quality within the limited processing budget available on mobile platforms. Lastly, every newly captured image provides an opportunity to refine the current reconstruction. As we will show later in this paper, the naive approach of repeatedly executing a batch-based reconstruction algorithm on the images in a first-in-first-out buffer leads to significantly degraded reconstruction quality.

Notably, the earliest DNN architectures for 3D reconstruction used recurrent neural networks (RNNs), for instance, 3D-R2N2 [6] and LSM [18]. However, the primary motivation for the recurrent nature was not the use case of iterative 3D reconstruction, but rather to allow a network to identify the features from a new viewpoint most complementary to previously processed viewpoints. These early recurrent architectures were soon superseded in terms of reconstruction quality by feedforward convolutional neural networks, such as Pix2Vox [8]. The main objective of the research presented in this paper was therefore to investigate whether the superior reconstruction of feedforward algorithms can be reconciled with the particular challenges of iterative 3D reconstruction. The key contributions presented in this paper are:An extension of the feedforward Pix2Vox+/F architecture [9] that supports iterative refinement.Qualitative and quantitative results that demonstrate the performance degradation of applying existing reconstruction algorithms in iterative settings both to the R2N2 dataset and to a higher-resolution extend version.A characterization of the variation in reconstruction quality that arises from viewpoint selection.

The remainder of this paper is structured as follows. In the next section, we discuss related work on 3D reconstruction. This is followed by Section 3, where we present the models that were tested in this study, our extension to the R2N2 dataset and define performance metrics for iterative 3D reconstruction. In Section 4, we benchmark our model with existing algorithms and evaluate the reconstruction quality of various viewpoint-selection algorithms. Lastly, Section 5 concludes the paper.

## 2. Related Work

Conventional monocular methods for 3D reconstruction, such as structure from motion (SfM) [19] and visual simultaneous localization and mapping [20] build a 3D map by estimating the camera position from corresponding features in different viewpoints. However, in a real-world environment, feature matching can be complicated due to insufficient overlap in consecutive images caused by artifacts and motion blur leading to unusable images. More recent methods try to solve this problem by training DNNs on large datasets of 3D objects. The key challenge in DNN-based reconstruction is to extract the most relevant features from each individual viewpoint and fuse these into a global reconstruction.

Multi-view stereopsis (MVS) refers to 3D reconstruction from calibrated overlapping images. Early data-driven MVS techniques include SurfaceNet [21,22], which combines all pixel and camera information into colored voxel cubes used as input of the network, and LSM [18], which uses differentiable projection for end-to-end training. MVSNet [23,24] infers the depth maps for each image one at a time using multiple source images and uses differentiable homography warping to enable end-to-end training. These methods have leveraged DNNs to obtain high-quality reconstructions, and recent works, such as R-MVSNet [25], put the focus on scalability. These MVS approaches, however, assume known camera parameters, which are generally obtained by a SfM approach. Due to this additional computation, these techniques are less suited for online applications.

One of the earlier multi-view 3D reconstruction approaches not using camera parameters was 3D-R2N2 [6]. Similarly to LSM [18], it uses an RNN to incrementally refine the latent code using feature maps from different viewpoints extracted by an encoder network. However, as stated by the authors of Pix2Vox++ [8,9], aggregating features before the decoder is very challenging. The Pix2vox++ model circumvents this complexity by producing separate reconstructions for each input image using an encoder–decoder network. A tertiary network then selects the most informative features from each single-view reconstruction. To fuse multiple single-view reconstructions, they are weighed together, preventing iterative updating.

The latest models have incorporated an attention-based mechanism to select relevant features from each input view. AttSets [7] is a CNN-based auto-encoder architecture. An attention module calculates the weights for each of the encoded images. The reconstruction is decoded from the weighted sum of latent codes. Transformer models incorporating self-attention have also been proposed for 3D reconstruction [10,11,12]. None of the attention-based methods supports iterative updating of a previous reconstruction, since these architectures expect to receive all input images at once.

Recently, researchers have started exploring neural implicit functions as an effective 3D geometry representation, representing 3D content in a continuous function in contrast with the traditional discretized voxel-based representations. Implicit functions such as neural radiance fields do not require 3D object representations during training and can be trained only on 2D images [13,26]. Neural implicit functions need to be retrained for every scene, making them in their current stage of development suited for applications that require online 3D reconstruction.

## 3. Materials and Methods

In this section, we first detail the architecture of our 3D multi-view iterative reconstruction model. Secondly we describe the performance metrics used in this paper. Lastly, we describe the original dataset and the extension we applied to create more viewpoints of each object in order to study the effect of viewpoint selection.

### 3.1. Iterative 3D Reconstruction Model

Our architecture for iterative 3D reconstruction is an adapted version of Pix2Vox++, see Figure 2, a state-of-the-art volumetric multi-view 3D reconstruction DNN [9]. The original model estimates a voxel representation of the 3D shape from a batch of 2D images. Our approach, however, processes a single frame at a time where each sequential frame is used to improve upon the 3D reconstruction generated based on the previous frames. We adapted Pix2Vox++/F, see Figure 3, a variant of Pix2Vox++ with a smaller computational footprint. Our adaptation will also work on the Pix2Vox++/A model.

The Pix2Vox++/F architecture is structured in three components: encoder, decoder and merger. A batch of 2D images is converted by the encoder into a lower-dimensional feature map. The encoder consists of a backbone model (we tested both VGG and ResNet backbones) followed by three blocks of a 2D convolution layer, 2D batch normalization and ReLU activation. The middle block has an additional 2D maxpool layer. The encoded features are then mapped by the decoder to a 3D voxel representation per input view. The decoder architecture consists of four blocks each having a 3D transposed convolution layer, a 3D batch normalization layer and a ReLU activation. This is followed by a fifth block with a last 3D transposed convolution and a sigmoid activation function.

Due to self-occlusion, different viewpoints offer better visibility for certain parts. The goal of the merger is to select the best parts in each of the individual 3D representations produced by the decoder. The merger outputs a score map for each volume generated by the decoder, which are normalized using the softmax function. Its architecture comprises six blocks of a 3D convolutional layer, 3D batch normalization and a leaky ReLU activation. The outputs of the first four layers are concatenated and used as input for the fifth layer. To address the loss of detail in the feature maps of the deepest convolutional layers of the merger, an additional tensor is provided as input to the merger containing the feature maps of the last ReLU activation and 3D transposed convolutional layer of the decoder. This tensor contains nine channels per voxel. The final (single) 3D reconstruction is obtained by calculating a weighted sum of these volumes according to these score maps.

In an iterative setting, not all input images are available at once. For the encoder and decoder model, this is not an issue, since their outputs do not depend on other images. The merger, however, as discussed above, uses softmax normalization to normalize the score maps for each image in the batch. This prevents its use in an iterative setting. By keeping a fused context tensor in memory, see Figure 3, the need for all context tensors at once is reduced to two: the fused context and the context of the incoming image. To keep the fused context tensor up to date at each new iteration, we repeated the score maps created by the merger nine times. A new fused context can then be obtained by calculating a weighted sum of the new and fused context according to the repeated score maps. From the fused context, the fused reconstruction can be obtained by taking the last channel of the tensor.

Even though we did not alter the architecture of the merger, its input changed, which resulted in a different training regime. The merger now has to learn how to compress all previously seen information into one fused context tensor instead of solely trying to combine the best parts of each reconstruction.

### 3.2. Performance Metric

The reconstruction model assigns a probability score to each voxel in the 3D representation. When this score exceeds a set threshold value, the voxel is assumed to be occupied. In this work, we used a threshold value of 0.4. This value was determined after an initial hyperparameter search; see Appendix B.

The quality of a single 3D reconstruction was measured by comparing the set of voxels in the reconstructed representation *A* with the set of voxels in the ground truth *B*. In particular, we used the intersection over union (IoU) ratio:(1)$$\mathrm{IoU}\left(A,B\right)=\frac{A\cap B}{A\cup B}$$

We assume that new 2D viewpoints become available at each discrete time step t=1,2,…,N. Each viewpoint provides an opportunity to update the 3D reconstruction. We denote IoU(t) as the IoU of the reconstruction obtained after the first t viewpoints have become available. When the viewpoint at time t is not selected for processing, IoU(t) = IoU(t − 1). Some example IoU trajectories for different selection algorithms are shown in Figure 4. Since our interest is to have high reconstruction quality at each time step, our performance metric is the average IoU over time, corresponding to the area under the curve:(2)$$\bar{\mathrm{IoU}}=\frac{1}{N}\sum_{t=1}^{N}\mathrm{IoU}\left(t\right)$$

### 3.3. Datasets

#### 3.3.1. ShapeNet

R2N2 [6] is the most widely used benchmark for deep-learning based multi-view 3D reconstruction algorithms. The dataset contains 32 × 32 × 32 voxelized representations of objects in 13 categories. These 13 categories are a subset of the larger ShapeNet [27] dataset. For each object, 24 images of resolution 137 × 137 are available with a view of the object. Viewpoint coordinates are expressed in spherical coordinates: distance, azimuth and elevation. Viewpoint azimuth values were randomly sampled over the full domain interval of [0,2π), but elevation and distance were sampled from limited ranges.

To allow for an in-depth analysis of viewpoint-selection algorithms, we require more than 24 viewpoints. For this reason, we created additional renderings (2D images) from the original ShapeNet model using the Pytorch3D library [28]. We used the same voxel representations for each object as the R2N2 benchmark. Compared to the 137 × 137 R2N2 resolution, we rendered the images at 224 × 224 resolution, since the pretrained VGG backbone takes this resolution as input. Some examples are shown in Figure 5.

Per object, 120 viewpoints were rendered according to a simulated drone flight path. Subsequent viewpoints have a fixed azimuth difference of $3^{\circ }$, and distance and elevation vary according to two sine waves with random phases.

#### 3.3.2. ABO Dataset

To demonstrate the generalizability of the models, we also experimented with the Amazon Berkeley Objects (ABO) dataset [29]. The ABO dataset is a recent large-scale dataset for real-world 3D objects. Each sample consists of a high-resolution 3D model accompanied by 30 renderings at a resolution of 1024 × 1024. The dataset also provides class labels for each sample, some of which align with the classes used in the R2N2 rendering discussed above. We created ground truths by converting the 3D model files to binary 3D voxel grids [30,31]. We obtained a subset from the ABO dataset authors containing 400 samples; filenames of the objects will be available upon request.

## 4. Results

In this section, we first benchmark our iterative 3D reconstruction algorithm to existing algorithms in the literature by the IoU metric (Equation (Equation 1)). In Section 4.3, we present an evaluation in terms of the instantaneous IoU(t) and demonstrate the problems that arise by naively deploying batch-based algorithms in an online setting. In Section 4.4, we empirically evaluate the variance in reconstruction quality in terms of the selected viewpoints and the order in which these are presented. Lastly, in Section 4.5, we compare the reconstruction quality obtained from different selection algorithms.

### 4.1. Final Reconstruction Quality

In our first experiment, we compared the quality of 3D reconstructions obtained by our iterative approach to the reported performances of other state-of-the-art approaches on the R2N2 dataset, as listed in Table 1. In particular, we evaluated the IoU obtained after processing N=8 views. For the batch-based algorithms, we report the IoU of the (only) reconstruction obtained from batched input. For the iterative algorithms, we report the IoU of the last reconstruction when sequentially processing k=8 images.

From these results, we can conclude that our iterative model results in better reconstructions than the RNN-based 3D-R2N2 model, which is to our knowledge the best-performing iterative model (aside from LSM [18], but this model requires camera parameters as input). We can also conclude that most batch-based reconstruction methods outperform iterative reconstruction. However, there is only a slight degradation in reconstruction quality of our model compared to the Pix2Vox++/F architecture from which it was derived. This slight decrease can be attributed to the increase in the difficulty of the problem because the iterative model also has to learn to compress the previously seen viewpoints.

In Table 2, we provide per-class IoU values for our algorithm and the original Pix2Vox++/F architecture. We report results on the original and the extended version of the R2N2 dataset. As the Pix2Vox++/F paper only reports overall IoU, and the weights of the trained model are not publicly available, we retrained the Pix2Vox++/F model, achieving an overall IoU of 69.6% (whereas the original paper reported 69.0%). We provide results for two different backbone models: VGG and ResNet. We obtained similar results as those in [9]: ResNet outperformed VGG by 1–2% IoU on average.

Table 2 reveals wide variation in the per-class IoU both on the original and on the extended dataset. The difference in per-class IoU between batch-based and iterative reconstruction was limited to ∼2%.

The most notable result of Table 2 is that both algorithms achieved lower reconstruction quality on the extended dataset. We attribute this performance loss to the smaller viewpoint diversity in this dataset. In the original dataset, the azimuth of 24 viewpoints is uniformly distributed in the interval [0,2π). Any selection of eight viewpoints is likely to fall within a large azimuth range. Selecting eight viewpoints out of 120 may result in batches covering much smaller azimuth ranges. In the worst case, eight consecutive viewpoints are selected, covering an azimuth interval of $8\times 3^{\circ }$. We investigate the importance of selecting diverse viewpoints further in Section 4.5.

### 4.2. Generalizability

To test the generalizability of our approach, we applied the model trained on the ShapeNet dataset to data from the ABO dataset. Table 3 provides per-class IoU values for our algorithm and the original Pix2Vox++/F architecture. Note that the model was not retrained on this new dataset. We obtained similar results as in the previous section. Our iterative approach obtained comparable but slightly lower IoU scores than the baseline approach without relying on batch input. We also included four classes which the model did not see in examples during training. We can observe that the models are still able to approximate 3D models, with the hardest classes being the rug and pillow class. These model classes share the least similarity with the classes on which the model was trained.

### 4.3. Online Reconstruction

In this section, we focus on the online reconstruction quality (IoU(t)) and its computational requirements. We compare our iterative algorithm to an online setup with the original Pix2Vox++/F model. The latter model can take as input batches of any size, including batch size 1 (single-viewpoint). Using the Pix2Vox++/F model in an online setting can therefore be achieved by buffering the most recent viewpoints, using a first-in-first-out replacement strategy, and using this buffer as batch input to the algorithm.

In our initial experiments, we presented a new viewpoint every timestep from the extended dataset, in order of increasing azimuth. For both algorithms, we observed a high fluctuation in reconstruction quality; see Figure 6. For Pix2Vox++/F, this can be explained by the fact that reconstructions are always based on a batch of consecutive viewpoints with minor difference in camera perspective. Although our model is equipped with a fused representation containing all information from previous problems, we still observed long-term memory loss. This phenomenon that features extracted from initial inputs tend to be forgotten as more viewpoints are processed is a long standing problem in machine learning [32].

To avoid long-term memory loss in the iterative algorithm and ensure sufficient viewpoint diversity in the input batch of Pix2Vox++/F, we therefore conducted our benchmark experiment on a subsampled version of the original R2N2 dataset. After ordering the 24 viewpoints of an object by azimuth, we selected every second image.

Table 4 compares the average IoU over time ($\bar{\mathrm{IoU}}$) when presenting the viewpoints ordered by azimuth. We experimented with several batch-composition strategies. The condition Size = 1 refers to a batch size of 1, with which a new reconstruction is calculated each time a new viewpoint becomes available, which is the same as single-view reconstruction. In the other conditions, a batch size of 3 viewpoints was used, which was updated with a stride of 1 or 3; see Figure 7 for additional clarification. Stride 1 corresponds with a first-in first-out buffer update strategy, whereas stride 3 corresponds with erasing the buffer after calculating a reconstruction. As a reference, we added the condition with a growing buffer, where each time a reconstruction is created based on all previously seen information. This last condition, however, has to redo a lot of computations every time a new viewpoint is added to the buffer. This results in the computation time rising quadratically with the size of the batch and is thus not feasible in practice.

Limiting the batch size of Pix2Vox++ induces a performance degradation in the reconstruction quality. Only the condition where all viewpoints in the batch are processed every timestep (Growing Buffer) results in a performance that is on-par with our algorithm. This setting, however, in addition to being slow, also requires keeping 12 images in memory, which results in additional memory resources when compared to the fused-context tensor from the iterative approach.

To quantify the speed of both reconstruction models, we testde them on various hardware setups, as shown in Table 5. We include results on a Jetson TX1 to show their performances on an embedded system. As different conditions produce different numbers of reconstructions (NoR), we compare the processing time for an entire sequence. We show that only the batch conditions with Size = 1, Stride = 1 and Size = 3, Stride = 3 are faster than the iterative approach, but they result in a loss in IoU.

### 4.4. Viewpoint Selection

The results presented in the previous sections already hinted to the importance of only processing a subset of the captured viewpoints. In this section, we compare different heuristics to determine what frames to process. In addition to improving the overall reconstruction, this also reduces the computational footprint of the reconstruction algorithm, since this scales linearly with the number of images processed. In the next section, we study several selection algorithms. We characterize the difference in IoU when a fixed number of *k* viewpoints is selected from the *n* images available. Since selecting *k* elements out of *n* scales exponentially in *n*, we only present quantative results obtained from testing all possible combinations of the smaller R2N2 dataset (n=24). For the larger extended dataset (n=120), we only report qualitative results. In order to save on computation, we performed the experiments in batches with the original Pix2Vox++/F model.

Table 6 shows different percentiles P, minima and maxima and the P95–P05 interpercentile range of the IoU obtained over all possible combinations of 3 out of 24 images (2024 combinations per object). Notably, a quarter of the tested combinations of three viewpoints resulted in an IoU that is on par with or better than the results presented in Table 2 for eight viewpoints. The results also highlight the importance of view selection, since there is on average a 16.3% variation between the best and worst combinations (95th and 5th percentile to avoid outliers). Worth noting is that the reconstruction of some classes is more robust to the selected viewpoints than others—the car class being the most robust.

To better understand which viewpoints contribute more to the reconstruction quality, we extended our study to the extended version of the dataset. Since there are 280,840 combinations of 3 out of 120 viewpoints per object, an exhaustive study is intractable. Figure 8 shows qualitative results of four objects. More examples can be found in Appendix C.

In this figure, the histograms of IoU values confirm the impact of viewpoint selection in obtaining good reconstructions: there are differences up to ∼25% between best and worst IoU for several objects. The second row in each subfigure indicates the distance, elevation and azimuth of each viewpoint. The red and green curve indicate how many of the 119×118 combinations, including that viewpoint, belong to the overall best 25% and worst 25%, respectively. These graphs reveal that each object has clear azimuth intervals with informative and uninformative viewpoints. We will revisit this question later in this section.

To create the bottom graph of each subfigure, we binned all combinations by viewpoint diversity and calculated the average IoU of each bin. We define viewpoint diversity as the angle of the shortest arc on the unit circle that contains the three points corresponding to the azimuth values. More details can be found in Appendix A. Again, these results indicate the importance of selecting sufficiently good viewpoints. A clear degradation of the IoU can be observed for a viewpoint diversity of 180, meaning that two of the three viewpoints in the tested combination were opposite viewpoints. All objects in the R2N2 dataset have at least one symmetry plane, and two viewpoints containing opposite viewpoints on the axis orthogonal to that plane contain only redundant information.

The distinct intervals in Figure 8 with informative and uninformative viewpoints raise the question of whether these intervals are linked to the object’s characteristics. Moreover, in real-world deployments, iterative reconstruction can start from any viewpoint. Starting the iterative reconstruction in a degraded region does not necessarily jeopardize the final reconstruction quality, as subsequent viewpoints can compensate for the missing information, but the IoU(t) will be degraded during the initial phase. Therefore, we calculated the reconstruction quality obtained from each individual viewpoint. Results for eight objects are shown in the polar plots of Figure 9, and more examples can be found in Appendix D. In this figure, the radial axis reflects the reconstruction quality, and the angular coordinate corresponds to the viewpoint azimuth.

In most polar plots, a butterfly-shaped curve is observed, most distinctively for instances that have one characteristic orientation, such as cars, telephones and aeroplanes. Viewpoints taken from perspectives aligned or orthogonal to that orientation tend to be less favorable starting points for iterative reconstruction. Somewhat surprisingly, the single-view reconstruction quality fluctuates even for objects with rotational symmetry, such as the lamp. However, one should also account for the fluctuating distance and elevation of subsequent viewpoints; see Figure 8. For the lamp, the viewpoints most informative to the reconstruction are those taken from perspectives with lower elevation, because these perspectives allow for a better estimation of the height of the lamp. The second row in Figure 9 shows four telephone examples with very similar 3D models, yet the reconstruction quality varies significantly for similar azimuth angles. This indicates that the optimal viewpoint selection for one object does not necessarily guarantee good performance on other similar objects.

Combining all these results, we conclude that the primary goal of a viewpoint-selection algorithm is to maintain sufficient diversity in azimuth and elevation. Limited benefits can be expected from applying more advanced strategies based on particular characteristics of the object to be reconstructed. Another argument in favor of simpler strategies is that no prior assumptions can be made with respect to the azimuth of the starting viewpoint. By applying the principle of Occam’s razor, we study in the next section only sampling strategies that are agnostic to particular object characteristics.

### 4.5. Reconstruction with Selected Viewpoints

We tested four viewpoint selection methods on our extended dataset. These methods are illustrated in Figure 4 and explained below. For a fair comparison, the parameters of each method were tuned such that each method selected n=8 viewpoints out of N=120.

•Random selection samples *n* had indices in the interval [1,N] without replacement; each viewpoint had an equal chance to be added by the selection. This is the standard way of selecting viewpoints for batch-based reconstruction methods [6,9].•Periodic selects the viewpoints with indices 0,T,2T,…,(n−1)T with $T=\frac{N}{n}$.•Rampup selects viewpoints with increasing intervals between subsequent indices. The rationale behind this method is to sample more viewpoints with lower indices in the ordered set, such that a higher IoUt(t) can be obtained in the initial phase of sequential reconstruction. We implemented this method using the following equation:
(3)
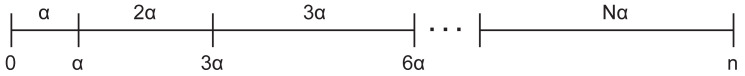

(4)α+2α+3α+…+Nα=n
(5)$$\alpha =\frac{2n}{N(N+1)}$$•MSE difference calculates the mean-squared error (MSE) (pixel-wise difference) between the current viewpoint and the previously selected viewpoint. This MSE is then multiplied by an arbitrary value; the resulting value is then the chance of selecting that viewpoint. This strategy was chosen in favor of handpicking a specific threshold because for different objects, the baseline MSE between consecutive viewpoints can vary substantially. The arbitrary value was then chosen such that for most objects, n=8 viewpoints were selected. The assumption was that MSE differences are correlated with the amount of additional information in the candidate viewpoint. Figure 10 shows an example trace.

Table 7 shows the results of the four selection techniques tested. While all selection strategies obtained a comparable IoU, the periodic and ramp-up strategies achieve higher $\bar{\mathrm{IoU}}$. According to these results, randomly selecting viewpoints does not guarantee sufficient viewpoint diversity, and the MSE difference is not a good predictor of the additional information contained in a viewpoint. Selecting more of the initial viewpoints does not result in a significant improvement over periodic sampling in terms of $\bar{\mathrm{IoU}}$. We attribute this observation to the existence of longer azimuth intervals with less informative viewpoints; see Figure 8. When coincidentally starting the iterative reconstruction in such a region, fewer informative viewpoints will be selected.

## 5. Conclusions

In this paper, we studied the problem of monocular, iterative, multi-view 3D reconstruction. Compared to the traditional batch-based 3D reconstruction, iterative reconstruction entails novel challenges related to viewpoint selection for maximizing IoU over time with the minimum number of images possible. We showed that existing batch-based 3D reconstruction methods, such as Pix2Vox++, can be modified to work in an iterative setting. Our method, based on iteratively updating both the context vector and the reconstruction, outperforms repeated batch-based processing, since it counteracts possible long-term memory loss while being much more computationally efficient. We qualitatively and quantitatively studied the effects of viewpoint diversity and selection on reconstruction quality. Results indicate distinct intervals with less and more informative viewpoints, but these intervals are highly object-specific. This makes creating a more intelligent selection system infeasible with the current state of 3D deep learning models. We can thus conclude that periodic selection of viewpoints generally provides the best results in terms of in-flight and final reconstruction quality. In future work, we will investigate if this object specificity is related to the low resolution of the 3D reconstruction. We will research deep learning approaches that can produce higher-resolution 3D reconstructions without increasing computational requirements.

## Figures and Tables

**Figure 1 sensors-22-09782-f001:**
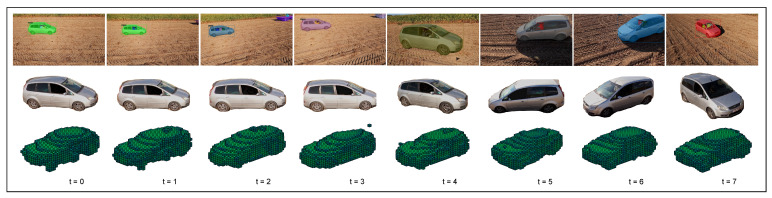
This figure illustrates the main idea of iteratively refining a 3D reconstruction. We generated a 3D reconstruction with the first available image. The next images were then used to improve upon this reconstruction using information from other viewpoints. The color of each voxel indicates the confidence of the model: green and blue represent high and low confidence, respectively. The video clip was captured with a GoPro carried by a Cinelog35 FPV drone. We then extracted the car from the images using an off-the-shelf segmentation model [17]. The reconstructions were obtained with the iterative model discussed in this paper.

**Figure 2 sensors-22-09782-f002:**
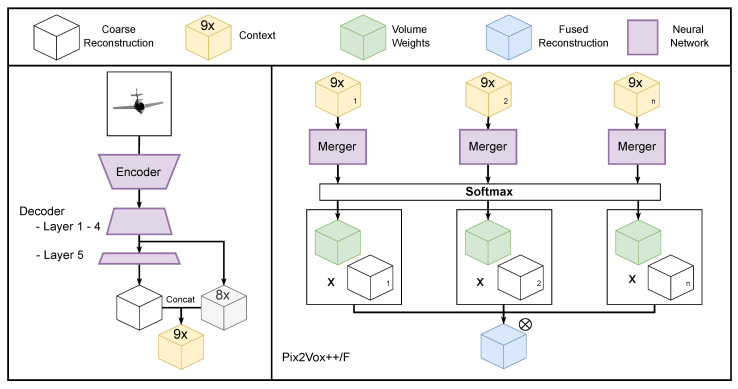
In the original Pix2Vox++/F architecture, the merger receives a context tensor for each input image and produces a single fused reconstruction. To maintain sufficient detail, this context tensor is a concatenation of the feature maps outputted by the 4th layer of the decoder and the coarse 3D reconstruction outputted by the final layer of the decoder.

**Figure 3 sensors-22-09782-f003:**
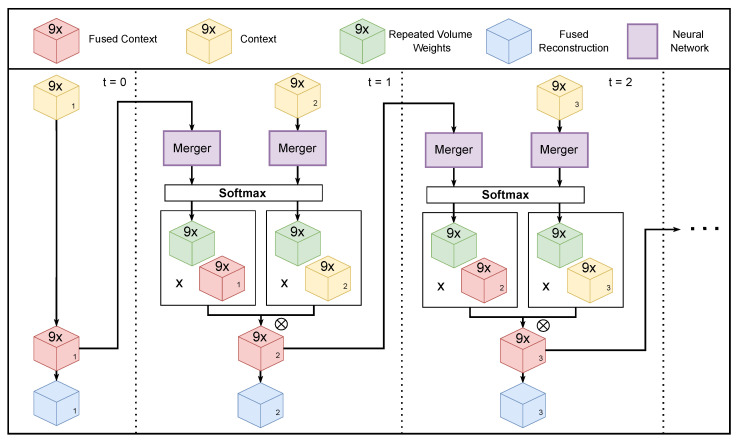
Adaptation for iterative 3D reconstruction presented in this work. We gradually refine a fused reconstruction maintained in memory. When a new input image becomes available, a new context tensor is produced by an encoder–decoder block, as shown in Figure 2. The updated fused context tensor is obtained as a weighted sum of the new context tensor and the previous fused context tensor. Whereas Pix2Vox only merges coarse reconstructions, we merge complete context tensors. To account for the additional channels during the merging, the score maps are repeated 9 times.

**Figure 4 sensors-22-09782-f004:**
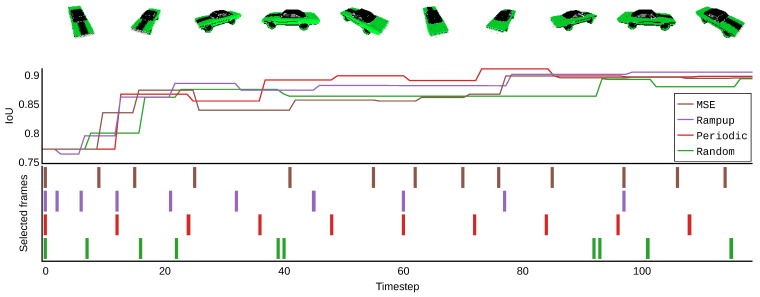
Principle of online, iterative 3D reconstruction. At each discrete timestep, an image is captured from a viewpoint that is in the immediate neighborhood of the previous viewpoint. When the viewpoint is selected, an updated reconstruction is calculated. The quality of this reconstruction is measured by the intersection over the union (IoU) of the produced reconstruction with the ground truth of the dataset. Different viewpoint-selection algorithms lead to different IoU curves, and not all images result in improved reconstruction quality. The selection methods are further explained in Section 4.5.

**Figure 5 sensors-22-09782-f005:**
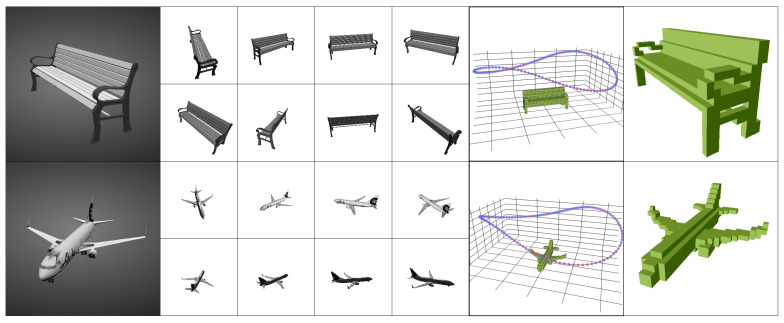
Two examples of our new rendering of the ShapeNet dataset. From the left, we first show the models as displayed on the taxonomy viewer on the ShapeNet website; next, we show eight of its input renderings. Lastly, we show the path along which the images were taken and the ground-truth voxels.

**Figure 6 sensors-22-09782-f006:**
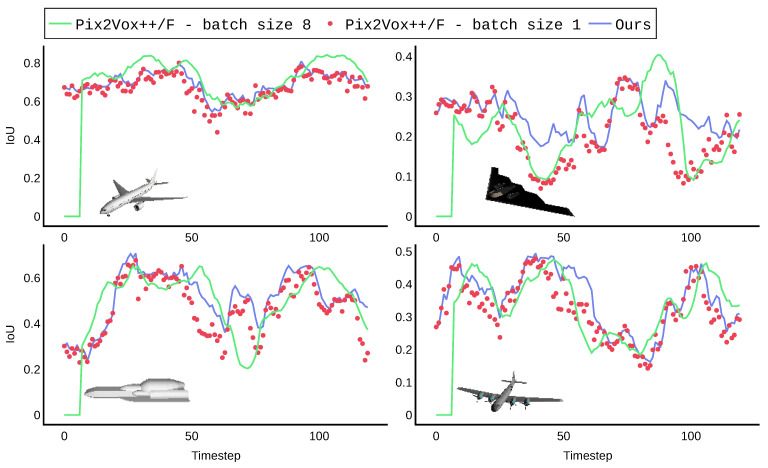
Reconstruction quality fluctuates significantly when processing all 120 images of the extended dataset in order of increasing azimuth. These fluctuations can be attributed for Pix2Vox++/F to the lack of diversity in the input batch, and for our iterative algorithm to long-term memory loss.

**Figure 7 sensors-22-09782-f007:**
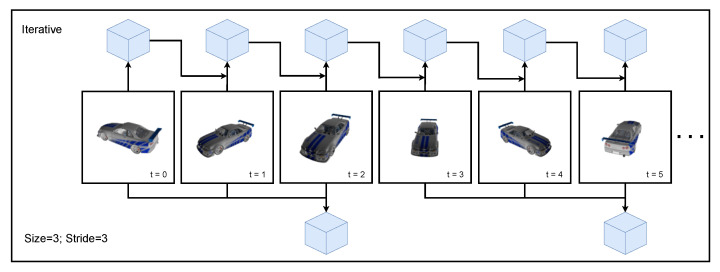
This figure gives an example of a batch processing method versus iterative processing. The iterative method is able to deliver a newly updated reconstruction every iteration (number of reconstructions (NoR) = 6), whereas the batch method produces two reconstructions (NoR = 2).

**Figure 8 sensors-22-09782-f008:**
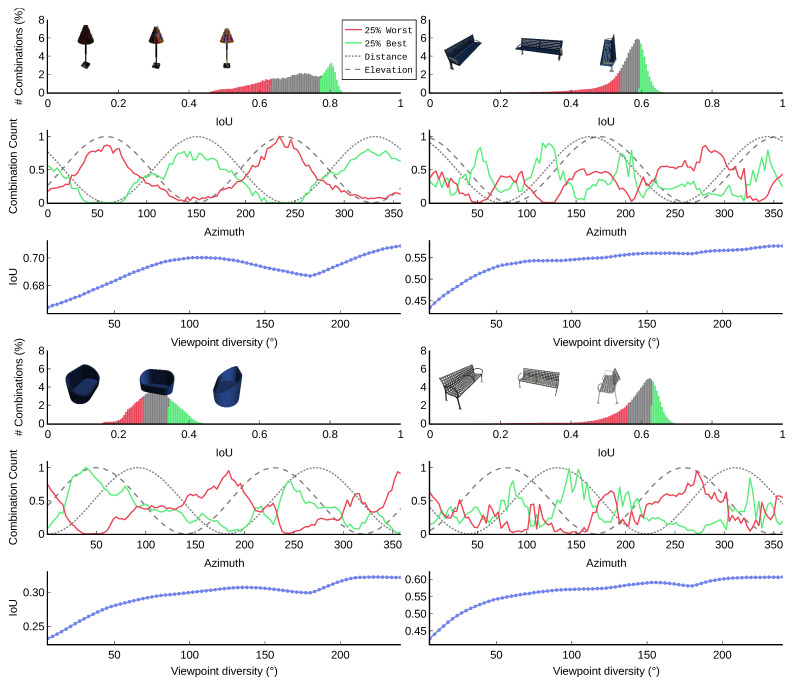
Qualitative insights on how viewpoint selection affects 3D reconstruction quality. For four different objects, all combinations of 3 out of 120 viewpoints were evaluated. For each object, the upper graph shows the histogram of IoU values, with red bars indicating the worst 25% selections and green bars indicating the best 25% selections. The middle graph indicates for each viewpoint (azimuth value) how many of the tested combinations containing that viewpoint belong to the worst or best 25% selections. Dashed lines indicate distance and elevation of each viewpoint. The bottom graph plots the IoU in function of the maximum distance between any two of the three viewpoints. Larger maximum distances indicate more similar viewpoints. The figure is best viewed in color.

**Figure 9 sensors-22-09782-f009:**
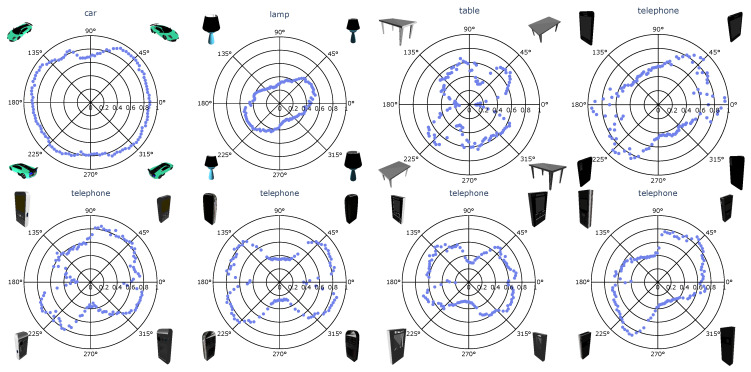
Polar plot indicating single-view reconstruction quality. IoU is shown on the radial axis, viewpoint azimuth on the angular axis.

**Figure 10 sensors-22-09782-f010:**
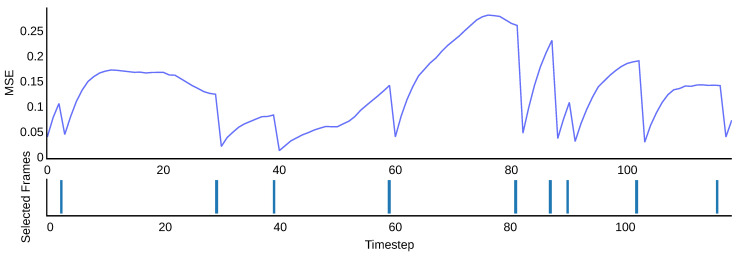
Example of selection using the MSE difference between the last selected viewpoint and the current viewpoint.

**Table 1 sensors-22-09782-t001:** Comparison of the IoU obtained on 8 randomly selected views from the R2N2 dataset. We take the mean for all objects in the test set. Performance numbers are taken from the referenced publications. The highest obtained IoU is highlighted in bold. The Iterative column indicates whether a method is capable of processing images iteratively. Our method achieves the best IoU when iterative processing is required.

Methods	IoU	Iterative
AttSets [7] (2020)	0.685	✗
Pix2Vox++/F [9] (2020)	0.696	✗
Pix2Vox++/A [9] (2020)	0.715	✗
EVolT [10] (2021)	0.698	✗
TMVNet [12] (2022)	**0.719**	✗
3D-R2N2 [6] (2016)	0.635	✓
Ours	0.690	✓

**Table 2 sensors-22-09782-t002:** Comparison of IoU per category, obtained from 8 views selected from the original or extended R2N2 dataset. We took the mean for all objects in the test set per category.

	R2N2				Extended R2N2			
**Category**	**Pix2Vox++/F**		**Ours**		**Pix2Vox++/F**		**Ours**	
**Backbone**	**VGG**	**Resnet**	**VGG**	**Resnet**	**VGG**	**Resnet**	**VGG**	**Resnet**
airplane	66.66	67.12	65.82	65.88	61.72	66.07	60.82	65.27
bench	58.92	61.66	57.63	60.91	56.99	60.52	55.54	59.13
cabinet	81.05	81.41	80.84	81.53	77.92	79.79	77.59	80.05
car	87.04	87.42	86.63	86.87	85.78	86.87	85.20	86.50
chair	60.92	62.61	59.84	61.78	58.97	61.54	57.48	60.82
display	58.00	61.36	58.34	61.11	57.42	60.73	56.32	58.95
lamp	46.99	47.08	46.21	46.61	44.67	45.08	43.30	44.93
speaker	72.45	74.43	72.56	75.09	72.75	73.12	72.10	74.09
rifle	63.96	64.83	63.34	63.97	63.29	64.48	61.86	63.56
sofa	75.13	75.85	75.26	75.44	72.88	75.31	71.81	75.41
table	63.15	64.75	62.59	64.23	60.28	63.59	59.60	62.95
telephone	79.27	85.03	77.58	84.58	81.92	83.40	80.13	83.48
watercraft	64.80	64.71	63.95	63.22	62.20	63.60	60.69	62.60
Overall	68.42	69.62	67.80	69.00	66.31	68.61	65.30	68.08

**Table 3 sensors-22-09782-t003:** Comparison of IoU per category, obtained from 8 views selected from the ABO dataset. We report the mean IoU over all objects per category.

Category	Pix2Vox++F	Ours
cabinet	41.1 ± 9.7	41.7 ± 8.3
chair	53.5 ± 2.7	49.4 ± 3.2
lamp	33.0 ± 4.7	36.0 ± 6.0
sofa	52.0 ± 3.6	50.6 ± 3.8
table	41.1 ± 5.9	39.6 ± 6.8
rug	27.4 ± 3.8	23.7 ± 3.9
pillow	36.4 ± 4.8	28.2 ± 6.7
bed	39.2 ± 5.7	47.1 ± 7.4
painting	37.8 ± 5.1	40.9 ± 4.8
ottoman	48.1 ± 8.6	55.8 ± 8.0
Overall	43.0 ± 1.7	42.4 ± 1.9

**Table 4 sensors-22-09782-t004:** Comparison of the average IoU over time obtained by each processing strategy on 12 selected viewpoints in the R2N2 dataset. The results show that our iterative method outperforms all batching methods in an online environment when resources are limited. We also display the number of reconstructions (NoR) during the sequence for each processing method.

	Iterative (Ours)	Batch (Pix2Vox++F)			
**Class**		**Size = 1**	**Size = 3**	**Size = 3**	**Growing**
		**Stride = 1**	**Stride = 3**	**Stride = 1**	**Buffer**
	**NoR = 12**	**NoR = 12**	**NoR = 4**	**NoR = 10**	**NoR = 12**
aeroplane	62.3 ± 1.1	50.7 ± 1.1	60.5 ± 1.2	61.0 ± 1.1	62.8 ± 1.1
bench	57.3 ± 1.5	41.8 ± 1.3	53.9 ± 1.5	53.8 ± 1.5	58.1 ± 1.5
cabinet	79.1 ± 1.9	70.5 ± 1.9	77.6 ± 1.9	77.7 ± 1.9	78.9 ± 1.9
car	85.0 ± 0.5	79.4 ± 0.5	84.7 ± 0.5	84.9 ± 0.5	85.4 ± 0.5
chair	58.1 ± 0.9	45.9 ± 0.8	55.2 ± 0.9	55.7 ± 0.9	58.6 ± 0.9
display	57.0 ± 2.5	44.1 ± 2.0	54.2 ± 2.3	54.9 ± 2.3	57.6 ± 2.4
lamp	45.5 ± 2.1	40.9 ± 2.1	44.8 ± 2.1	44.9 ± 2.1	45.7 ± 2.2
speaker	72.6 ± 2.2	65.4 ± 2.2	71.3 ± 2.2	71.4 ± 2.2	72.4 ± 2.2
rifle	61.1 ± 1.4	44.6 ± 1.1	57.7 ± 1.3	57.4 ± 1.3	61.0 ± 1.4
sofa	71.9 ± 1.2	59.4 ± 1.1	68.8 ± 1.2	69.1 ± 1.2	71.8 ± 1.2
table	62.4 ± 0.8	51.1 ± 0.7	60.0 ± 0.8	59.6 ± 0.8	62.8 ± 0.8
telephone	80.3 ± 2.3	66.9 ± 2.2	77.3 ± 2.4	78.5 ± 2.4	79.6 ± 2.4
watercraft	60.1 ± 1.5	48.5 ± 1.3	58.9 ± 1.5	58.8 ± 1.4	61.0 ± 1.5
Overall	66.3 ± 0.4	55.8 ± 0.4	64.3 ± 0.4	64.4 ± 0.4	66.6 ± 0.4

**Table 5 sensors-22-09782-t005:** Comparison of processing time for a single 3D reconstruction, reported in miliseconds and measured on various hardware platforms. Results are averages over reconstructions of 100 objects, with 20 images per object.

Processor	Iterative (Ours)	Batch (Pix2Vox++F)		
		**Size = 1**	**Size = 2**	**Size = 3**
Intel(R) Xeon(R)	248.7	149.5	293.9	437.8
i7-10850H	128.6	86.3	160.0	241.9
Jetson TX1	83.6	62.6	105.3	164.7
GeForce GTX 1080	11.1	8.5	13.1	17.9

**Table 6 sensors-22-09782-t006:** Percentiles P, minima and maxima of IoU values obtained from all combinations of 3 viewpoints. The last column indicates the difference between P05 and P95. One quarter of the combinations have similar or higher IoU than the results presented in Table 2 for 8 viewpoints. We also see a large variation between best and worst combinations.

Class	Minimum	P05	P25	P50	P75	P95	Maximum	P95 − P05
aeroplane	38.1	52.0	59.5	63.0	65.7	68.9	72.7	16.8
bench	29.1	43.6	51.6	56.2	60.1	64.6	69.7	21.0
cabinet	60.5	70.3	75.8	79.0	81.7	84.7	87.7	14.4
car	72.6	80.6	84.0	85.7	87.0	88.5	90.0	7.9
chair	37.2	48.3	54.5	58.1	61.2	64.8	69.3	16.6
display	31.3	41.5	50.5	56.5	61.6	67.3	73.5	25.8
lamp	31.8	38.1	42.6	45.7	48.4	51.9	56.0	13.8
speaker	56.7	64.2	69.3	72.5	75.3	78.4	81.6	14.2
rifle	35.4	48.4	55.7	59.9	63.5	67.8	73.1	19.4
sofa	49.5	61.1	67.2	70.9	74.0	77.6	81.7	16.5
table	38.0	49.8	56.8	61.1	65.0	69.4	74.1	19.6
telephone	49.2	61.8	74.3	81.6	86.0	89.7	92.4	27.9
watercraft	36.7	50.0	56.9	60.7	63.9	67.6	71.9	17.5
Overall	45.3	56.2	62.4	66.0	69.0	72.5	76.3	16.3

**Table 7 sensors-22-09782-t007:** Results for different selection techniques. We report both the IoU and the $\bar{\mathrm{IoU}}$ averaged over the full test set for 8-view reconstruction in %. We highlight in bold the highest achieved IoU and $\bar{\mathrm{IoU}}$ per category and overall.

	Random		MSE		Periodic		Rampup	
**Category**	**IoU**	$\bar{\mathrm{IoU}}$	**IoU**	$\bar{\mathrm{IoU}}$	**IoU**	$\bar{\mathrm{IoU}}$	**IoU**	$\bar{\mathrm{IoU}}$
aeroplane	65.12	60.45	64.62	59.21	**66.37**	62.09	66.13	**63.06**
bench	59.03	53.83	60.07	54.60	**60.76**	55.73	60.30	**56.43**
cabinet	79.91	76.09	80.39	76.53	**81.13**	77.22	81.10	**77.73**
car	86.40	83.55	86.63	83.93	**86.98**	84.40	86.95	**84.85**
chair	60.88	55.86	61.41	56.36	**62.68**	57.76	62.12	**58.15**
display	59.35	53.87	61.45	56.36	**60.77**	56.01	60.59	**56.29**
lamp	44.77	42.43	43.88	41.56	**45.16**	43.18	45.12	**43.48**
speaker	73.88	70.09	74.21	70.71	74.30	71.12	**74.47**	**71.70**
rifle	63.12	58.90	62.76	57.89	**64.19**	60.49	64.09	**61.27**
sofa	75.53	70.29	76.14	70.94	**76.87**	71.88	76.61	**72.44**
table	63.06	58.69	63.37	58.92	63.55	59.98	**63.67**	**61.01**
telephone	83.43	77.57	83.18	78.26	**84.40**	**79.09**	83.06	78.96
watercraft	62.78	58.16	63.12	57.90	**64.16**	60.20	63.86	**60.63**
Overall	68.06	63.81	68.31	63.96	**69.08**	65.23	68.90	**65.85**

## Data Availability

The R2N2 renderings and 3D voxel models are available at https://cvgl.stanford.edu/3d-r2n2/ accessed on 8 December 2022. The ShapeNet dataset is available at https://shapenet.org/ accessed on 8 December 2022. The ABO dataset is available at https://amazon-berkeley-objects.s3.amazonaws.com/index.html accessed on 8 December 2022. The filenames of the ABO subset used in this paper will be available upon request.

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
