# Peer review of "Iterative Online 3D Reconstruction from RGB Images"

_sensors, 2022, doi:10.3390/s22249782_

Round 1

Reviewer 1 Report

The paper presents an extension of pix2vox algorithm for the inference of 3D volumes from (a sequence of) 2D images. The authors propose an interactive vs batch approach and demonstrate its benefits by extensive experiments on a public dataset. The novelty of the work is minimal and maybe its merit is limited to an ablation study with the pix2vox algorithm with respect to viewpoint selection strategy. But it is not clear to me why not prefer a batch approach since no next best view selection is proposed.

The related work is not thorough enough. The authors should also refer to non-CNN techniques and discuss their shortcomings. Also CNN techniques such as [6] and [17] should be briefly presented and qualified with respect to the proposed work.

The description of the proposed extension is also not clear enough. I propose to display the architectures of original pix2vox and proposed side by side to illustrate the differences.

The experiments are well explained.

Reviewer 2 Report

This paper tried to build an iterative online 3D reconstruction algorithm which was based on Pix2Vox++/F[18]. Though the conversion from batch-based Pix2Vox++/F [18] to the proposed iterative algorithm is straight forward, the explanation of the proposed architecture in Fig. 2 is not clear. Authors spent less than one page (page 4) in introducing Fig.2, but much more pages (sections 4.1-4.2, nearly 4 pages) in related experiments that can be expected in advanced (in principle, iterative will be inferior to batch in performance). The concept from batch to iterative conversion is hence not clear (see below). The authors spent a lot of pages to introduce “viewpoint selection”. However, I cannot see its relation to “iterative” since after selection, the sampled viewpoints will not be adjacent. If so, does it still need to determine the presentation order to the iterative algorithm for these sampled viewpoints? If the time spent to determine the presentation order of the selected viewpoint is large, users would like to use the batch-based algorithm, but not the iterative algorithm, to get higher reconstruction quality. The application for the viewpoint selection (e.g., 3 out of 24, 8 out of 120) is not clear, which prevents its linkage with the main theme of this paper (i.e., iterative).

1.   Fig. 2. What is context is not explained. What is the architecture of “merger”? Is it a traditional computing block or another deep learning block? How to determine the weight by merger for fused reconstruction and context? The weights are constants (pre-determined) or determined by learning? In my opinion, if the merger is a traditional computing block, then the conversion from Pix2Vox++/F to iterative is straight-forward, but still needs to be well-explained.

2.   From Table 1, the proposed algorithm is slightly better than the 3D-R2N2 [6]. However, 3D-R2N2 was proposed in 2016 and Pix2Vox++/F was proposed in 2020. This comparison is unfair and the authors’ contribution or credit is not clear and enough.

3.   In Table 3, the batch style with (size, stride) is not clear. It should be illustrated with a simple figure. Is it meaningful to compare the iterative with the batch whose size is not 12?

Reviewer 3 Report

This study exploits an online 3D reconstruction approach and achieved promising experimental results. In general, the manuscript is well-written and rich in details. The adopted methodologies, the description of the models, and the obtained results are well-described and comprehensive. The paper could benefit from some major improvements. Here are some suggestions to be considered.

1.  In literature review part, I suggest the author considering more existing works, such as the MVSNet series.

2.  I suggest that Figure 2 be rearranged. There exist too many arrows with different directions, which make the readers confusing at the first glance. Usually, the input is put at the top/left in a pipeline figure.

3.  The reason of VGG chosen as the backbone should be explained in this paper. According to Pix2Vox++ original implementation, the VGG16. VGG19, Resnet50 and the DenseNet could be the backbone for feature extraction. Why the VGG is chosen here?

4.   As the authors aim at reconstructing 3D voxels with high speed, I suggest that the inference time of each approach should be added in the manuscript.

5.  I suggest more experiments be conducted. Results on R2N2 and the R2N2 extended seem not enough. For example, experiments conducted on the Things3D dataset could be considered.

Round 2

Reviewer 1 Report

The authors have addressed my concerns in their revised manuscript

Reviewer 2 Report

This version has solved my problems in the previous review.